# Appeals to shared suffering in the context of the Israeli-Palestinian conflict

Lukas Reinhardt[1,2]*, Harvey Whitehouse[1]*

**1** Centre for the Study of Social Cohesion, School of Anthropology and Museum Ethnography, University of Oxford, Oxford, United Kingdom, **2** Identity and Conflict Lab, Yale University, New Haven, Connecticut, United States of America

* lukas.reinhardt@anthro.ox.ac.uk (LR); harvey.whitehouse@anthro.ox.ac.uk (HW)

## Abstract

Shared suffering can bond groups together and motivate fighting and even dying for the group leading to vicious cycles of violence while the suffering of the outgroup is routinely overlooked. But could the power of shared suffering also be harnessed to reduce tensions? Here we present evidence that a speech appealing to suffering shared by both sides in the Israeli-Palestinian conflict delivered by a professional politician in December 2023 can foster identity fusion, trust and openness to friendship towards the outgroup – Palestinians or Israelis – in a sample of Muslim and Jewish Americans (n = 159). Effects partially persisted three days after exposure to the speech. In a follow-up study we found that the same speech also increased positive attitudes towards Israelis and Palestinians in a sample of the general US population (n = 361).

## Introduction

After a long history of violence, the Israeli-Palestinian conflict escalated dramatically following the Hamas attacks of October 7, 2023. Since then, Israel has engaged in a military campaign that has killed tens of thousands of Gaza residents between October 2023 and June 2024, many of them children, and, at the time of writing, an unknown number after that period [1,2]. The International Criminal Court (ICC) Prosecutor Karim Khan KC has applied on 20 May, 2024 to the ICC for arrest warrants to be issued against members of the Hamas leadership as well as Israeli Prime Minister Benjamin Netanyahu and Israeli Defence Minister Yoav Gallant for war crimes and crimes against humanity (See https://www.icc-cpi.int/news/statement-icc-pros-ecutor-karim-aa-khan-kc-applications-arrest-warrants-situation-state [accessed 20 August, 2025]). On 21 November, 2024, the Court issued arrest warrants against Netanyahu and Gallant (See https://www.icc-cpi.int/news/situation-state-palestine-icc-pre-trial-chamber-i-rejects-state-israels-challenges [accessed 20 August, 2025]). In a case that was brought before the International Court of Justice (ICJ) on 29

provided the original author and source are credited.

**Data availability statement:** All relevant data are within the manuscript and its Supporting Information files.

**Funding:** Funded by an Advanced Grant ('Ritual Modes: Divergent modes of ritual, social cohesion, prosociality, and conflict', grant agreement no. 694986) from the European Research Council under the European Union's Horizon 2020 Research and Innovation Programme. (HW) This research was also supported by a Templeton Religion Trust Grant ("The Persistence of "Wild" Religious Traditions" grant agreement #TRT-2021-10490). (HW) The funders had no role in study design, data collection and analysis, decision to publish, or preparation of the manuscript.

**Competing interests:** The authors have declared that no competing interests exist.

December 2023, South Africa has alleged that Israel's military campaign is an act of genocide against Palestinians in Gaza (See www.icj-cij.org/case/192 [accessed 20 August, 2025]). The ICJ has also accused Israel of abusing its position as an occupying power and described Israel's presence in the Occupied Palestinian Territory as unlawful (See the ICJ advisory opinion from 19 July, 2024: https://www.icj-cij.org/sites/default/files/case-related/186/186-20240719-adv-01-00-en.pdf [accessed 20 August, 2025]).

What psychological changes could reduce the harms caused by conflicts of this kind? Although previous research suggests that perceptions of shared suffering with fellow group members can increase cohesion within the ingroup and fuel animosity towards the outgroup [3,4], the same psychological processes might also be harnessed to mitigate tensions if suffering is perceived to be shared by ingroup and outgroup [5]. In this article, we present two studies – one with Muslim and Jewish Americans and one with a sample from the general US population – that explore whether a speech appealing to suffering *shared by both sides* can improve attitudes towards the outgroup.

### Shared suffering in intergroup conflicts

One factor that contributes to the psychological dynamic of conflicts such as the Israeli-Palestinian conflict are perceptions of shared suffering of the ingroup which can create powerful group bonds (e.g., [6–10]) and motivate extreme pro-group action (e.g., [3,11,12]). At the same time, the outgroup is often de-humanised, and the suffering of its members concealed or ignored, fomenting further violence and cruelty (e.g., [13,14]) and hindering perspective taking (e.g., [15–20]). However, when Palestinians and Israeli Jews are reassured that their group's claims to victim status have been acknowledged, their stated willingness to reconcile increases [21]. The same study also found that members of the politically more powerful group – Israeli Jews – expressed greater willingness to take action towards resolving the conflict after being reassured that their group's victim status had been recognized.

Having one's grievances acknowledged may increase openness to reconciliation up to a point, but going further by recognizing that suffering is experienced on both sides of a conflict (i.e. by forming inclusive victim beliefs), could plausibly provide an even more potent motivation for peace-seeking (e.g., [22–26]). Inclusive victim beliefs may therefore provide an alternative perspective to competitive victim beliefs (e.g., [27–29]) and zero-sum thinking [30], perhaps providing a particularly effective pathway to the resolution of seemingly intractable conflicts.

Shnabel and colleagues [31] have demonstrated in an Israeli-Palestinian context that inducing a common victim identity through exposure to a text mimicking the style of a well-known newspaper about losses on both sides increased forgiveness. Acknowledging that both groups are victims can also be an important component of re-humanisation processes helping to re-discover further aspects of shared humanity [32–34]. Acknowledging shared suffering on both sides of conflict does not diminish or obscure claims to ingroup victimhood, thereby obviating the zero-sum logic associated with competitive victim beliefs.

## Present research

In this article we present two pre-registered studies that analyse the potential of appeals to suffering *shared by both sides* to improve attitudes towards the outgroup in the context of the Israeli-Palestinian conflict. We launched the two studies on 14 December 2023 and 15 December 2023, utilizing a recording of a speech that was written by us and designed to appeal to shared suffering on both sides. Studies that analyse effects of video interventions in polarized environments include Bailard and colleagues [19], Wuttke and colleagues [35] and Reinhardt and Whitehouse [36]. The speech was delivered by Lord John Alderdice who is a professional politician and former leader of the Alliance Party in Northern Ireland. We used the resulting video as a treatment in two online studies.

In the first study we recruited American participants of Muslim and Jewish faith, who identified with one side of the conflict and had been emotionally impacted by it. Due to the relatively small number of available participants of Jewish and Muslim faith, we chose a within-subjects design in which we measured attitudes towards Palestinians and Israelis before and after exposure to the speech. Compared to between-subjects designs, within-subjects designs increase power because they create two rather than one data point per participant and control for individual-specific effects by design [37]. In order to obtain the same power, between-subjects designs can require four times as many participants as within-subjects designs while this ratio can also be higher or lower depending on the parameters at hand [38]. Within-subject designs also allow us to obtain treatment effects on an individual level which facilitates the analysis of heterogeneous treatment effects even in relatively small samples.

Our outcome measures were identity fusion – a strong form of group bonding (see e.g., [39,40]) – as well as trust, and openness to friendship. As pre-registered, we combined these three variables into an index we refer to as "attitudes" that served as the main outcome variable. In the second study, we used the same design to investigate whether there would be similar effects for a sample of the general US population. To evaluate the durability of these effects, we reached out to all participants three days after the first wave of data collection and measured outcomes again. More than 90% of participants also took the second wave.

In December 2023 when we conducted our studies, the Israeli-Palestinian conflict dominated the news and public debate in the US. Thus, our studies allow us to assess the impact of appeals to suffering shared by both sides in a phase of the conflict in which the suffering of the ingroup was highly salient and the emotional impact of the conflict – especially on Muslim and Jewish participants – was strong. The emotional impact of the conflict was also exacerbated by a variety of content on social media which included horrifying images and videos as well as fake news and propaganda supporting both sides. These influences of social media played a substantially smaller role in earlier outbreaks of the conflict such as the 2014 Gaza war but will likely play an even larger role stoking emotional reactions to conflicts in the future. A key question is how different ways of portraying the sufferings of victims on both sides of a conflict may impact prospects for a peaceful resolution.

## Study 1

### Study 1 - design and hypotheses

Fig 1 includes a screenshot of the video of the speech that we use in the study, and we provide the full transcript of the speech in the next subsection. The video was 3 min and 16 seconds long and can be accessed via the following link: https://www.youtube.com/watch?v=181MWAiHIoo. The speech had three paragraphs. The opening and closing paragraphs of the speech were taken with slight modifications from a speech that Lord Alderdice delivered at the House of Lords (See Appendix S1 in S1 File). The main paragraph pertaining to shared suffering was written by us and was not used publicly before.

The opening paragraph acknowledged that it is very difficult to approach the debate over the situation in Israel and Gaza calmly. The main paragraph emphasized that suffering is shared by both sides. We illustrated this central argument

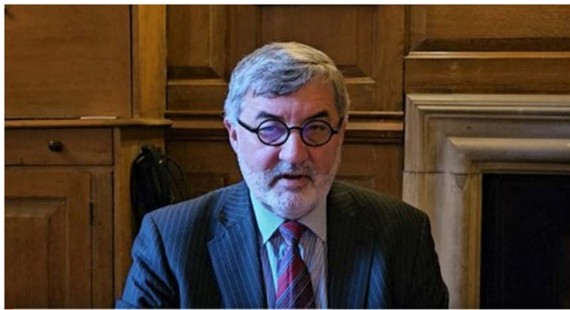

**Fig. 1. Screenshot of the video.**

by comparing the pain mothers and fathers from both sides feel when they see their children suffering. We also referred to the pain younger Israelis and Palestinians feel when they see their siblings suffer and we referred to the suffering of millions of people all over the world who observe the conflict from afar and might react with psychological trauma, depression, and despair. The closing paragraph urged leaders to intensify efforts to find solutions to the conflict arguing that working for lasting peace is the only way to build a better future for the region.

Lord Alderdice is a member of the House of Lords of the Parliament of the United Kingdom and a member of the Select Committee of the House on International Relations and Defence who has actively participated in the public debate about the Israeli-Palestinian conflict (see for instance his personal website: https://lordalderdice.com/index.php/category/articles/middle-east/). As a former leader of the Alliance Party of Northern Ireland he played a significant role in the Irish Peace Process. Lord Alderdice is also founding director of Oxford's Conference on the Resolution of Intractable Conflict. We selected Lord Alderdice as a speaker due to his substantial expertise on intractable conflicts. We acknowledge that a Northern Irish speaker might be perceived differently by a US audience than a US speaker. However, this feature of our setup allows us to analyse the impact of the speech per se without having complex interactions with partisan effects. Moreover, the fact that Lord Alderdice is likely not known by US participants mitigates the impacts of prior beliefs about the speaker on the effects of the speech.

We acknowledge that there were many minor idiosyncratic elements of our speech that went beyond the mere appeal that suffering is shared by both sides including specific wording choices, the personality of Lord John Alderdice, and the intonation used in the video. However, these framing aspects of the speech were necessary to create a treatment with high real-world relevance.

The procedure of our within-subject study was as follows. First, we obtained written informed consent. Then, we measured outcomes: i) identity fusion with Palestinians and Israelis, ii) trust towards a random Palestinian and a random Israeli, and iii) openness to friendship with a random Palestinian and a random Israeli. We randomized the order of the "attitudes towards Palestinians" and the "attitudes towards Israelis" blocks. Then, we measured demographics and conducted an attention check. Then, we showed participants the video and measured outcomes again. Finally, we asked a comprehension check question and asked whether any technical problems occurred. Appendix S6 in S1 File includes the online surveys we used.

We pre-registered that we use an index of fusion, trust, and openness to friendship as the main outcome variable. We normalised the values of all three single variables such that they ranged from 0 (unfavourable) to 1 (favourable) and used the average as the index value. Thus, every participant had four index values that captured attitudes towards Palestinians and Israelis before and after treatment. In the following we refer to index values as "overall attitudes" or just "attitudes".

We pre-registered that we will use t-tests for two main hypotheses for the first study. See AsPredicted #155,163: https://aspredicted.org/WPQ_MBX

H1: Participants will have better overall attitudes towards both Israelis and Palestinians after exposure to the speech.

H2: Individual level differences between overall attitudes to Israelis and overall attitudes to Palestinians will be smaller after exposure to the speech.

We will refer to results with $p < 0.05$ as "significant". We do not apply multiple hypotheses corrections as we focus on a small number of clearly defined pre-registered hypotheses. Our studies received ethics approval from the School of Anthropology and Museum Ethnography Research Ethics Committee at the University of Oxford. We obtained written informed consent from all participants. The individual pictured in Fig 1 has provided written informed consent (as outlined in PLOS consent form) to publish their image alongside the manuscript.

## Study 1 – transcript of the speech

The speech had the following transcript:

[Opening paragraph] My Lords, as the founding Director of Oxford's Conference on the Resolution of Intractable Conflict, I feel obliged to speak out about the terrible scenes we have witnessed in Israel and Gaza since the seventh of October. It is very difficult to approach this debate calmly. One might argue that there is no reason to be calm. Horrible, frightening things have been happening in Israel and Gaza and it is natural to rage and wish for vengeance and to identify more with one side of the conflict than the other.

[Main argument] But the truth is that the appalling sufferings of the current conflict have been felt by large numbers of innocent civilians on both sides. Many of us are parents ourselves who can surely imagine the extreme distress of both Israeli and Palestinian mothers and fathers who have seen their children mutilated and killed before their eyes. Or younger Palestinians and Israelis who have witnessed the traumatic and untimely deaths of their own siblings and who will be haunted by these ordeals for the rest of their lives. Please just reflect for a moment on the nature of such experiences. Are these the kinds of things that people can simply forget and forgive? And what about the millions of people throughout the world who weep in distress and burn with rage as they watch such things on their televisions? Every horror unfolding in the news right now is storing up decades of psychological trauma that for some will lead to depression or despair but for many others may set them on a path for revenge. But remember that a mother feels exactly the same when her child is in agony regardless of whether she is a Palestinian or an Israeli. If we just pause to consider how this horrible suffering is actually shared on both sides, maybe there is time to turn it into a force for good, a force that can convince us all to stop in our tracks, to fight back against the violence itself rather than to fight back against each other.

[Closing paragraph] Moving forward must mean finding a political settlement, my Lords. Is His Majesty's Government prepared to go beyond mere repetition of the wish for a '2-State Solution' and engage humbly and determinedly in a regional peace process that addresses the disturbed historic relationships, to which we have sadly contributed, and give hope in a place where it is in as short supply as food, water, and medicines, rather than provide and encourage the use of weapons as though there were not enough of them already in the region, and as though they were the solution, rather than part of the problem? My own experience is that those working for peace are often attacked by both sides, but working for peace is the only way to provide a future for our children and grandchildren.

## Study 1 - data

We ordered a sample size of 400 participants who were US citizens identifying as either Jews or Muslims at the online platform Prolific based on our experience with within-subjects designs. We did not conduct an a-priori power analysis because it seemed unlikely that we would be able to recruit the full $n = 400$ sample due to the limited number of available Jewish and Muslim participants registered on Prolific. Also, we considered $n = 400$ to be a sufficient upper bound given our prior experiences with within-subject designs. We collected data from December 14, 2023 until December 16, 2023 and we were able to collect data from 174 participants (see also Appendix S2 in S1 File which provides a timeline for all data collections.). As pre-registered, we excluded participants who failed the attention check, selected the wrong answer

in the comprehension check, or reported technical problems with the video, which left us with a final sample size of 159 participants. We paid our participants £1.05 which corresponded to an average hourly wage of £8.79. Forty-eight participants were Muslims, and 111 participants were Jews. Our primary measures were the Israeli-Index before the speech, the Israeli-Index after the speech, the Palestinian-Index before the speech, and the Palestinian-Index after the speech.

Participants were on average 42 years old, 76% had a university degree, 52% were females, and the average position on the political spectrum scale from 1 (left) to 10 (right) was 4.2. We also asked participants before exposure to the speech to what extent news reporting on violence between Israelis and Palestinians had impacted them psychologically with answers ranging from 'No impact' (1) to 'Extreme impact' (5). Participants reported an average psychological impact of 3.57 which ranged between 3 (Medium impact) and 4 (Strong impact). There were significant differences between Muslim and Jewish participants with regard to age ( $M_{Jews} = 46$; $M_{Muslims} = 33$; $p < 0.001$; two-tailed t test; $t(157) = -5.0011$), position on the political spectrum ( $M_{Jews} = 3.9$; $M_{Muslims} = 4.9$; $p = 0.039$; two-tailed t test; $t(157) = 2.0850$) and psychological impact ( $M_{Jews} = 3.44$; $M_{Muslims} = 3.85$; $p = 0.038$, two-tailed t test; $t(157) = 2.0850$) but not with regards to the share of university-educated participants ( $M_{Jews} = 0.78$; $M_{Muslims} = 0.71$; $p = 0.309$; two-tailed t test; $t(157) = -1.0210$) or the share of female participants ( $M_{Jews} = 0.49$; $M_{Muslims} = 0.58$; $p = 0.309$; two-tailed t test; $t(157) = 1.1192$). However, these differences are no primary concern since we do not focus on differences between Jewish and Muslim participants in our analyses.

## Study 1 - results

Participants had better attitudes towards Israelis after exposure to the video ( $M_{Index\ Israelis\ Pre\ Treatment} = 0.536$; $M_{Index\ Israelis\ Post\ Treatment} = 0.554$; $p = 0.002$; one-tailed t test; $t(158) = 2.867$; $d_z = 0.23$; H1). Participants also had better attitudes towards Palestinians after exposure to the video ( $M_{Index\ Palestinians\ Pre\ Treatment} = 0.443$; $M_{Index\ Palestinians\ Post\ Treatment} = 0.475$; $p < 0.001$; one-tailed t test; $t(158) = 5.002$; $d_z = 0.40$; H1).

Since our sample included supporters of both sides, we also analysed how the treatment affected attitudes towards the ingroup and outgroup. These analyses were exploratory. We determined the ingroup of participants by assessing to which side they had better attitudes before treatment. Seven participants had identical index values towards both sides before treatment and were excluded from all following analyses that focus on attitudes towards ingroup and outgroup rather than towards Israelis and Palestinians. As expected, most Muslims had better attitudes towards Palestinians (41) while only a minority had better attitudes towards Israelis (4). Moreover, most Jews had better attitudes towards Israelis (97) while only a minority had better attitudes towards Palestinians (10). Not surprisingly, religion correlated strongly with attitudes towards Palestinians and Israelis. However, these results also showed that religion and attitudes towards both groups did not correlate perfectly, reflecting the complex reality of the conflict.

There was no significant increase of attitudes towards the ingroup ( $M_{Index\ Ingroup\ Pre\ Treatment} = 0.732$; $M_{Index\ Ingroup\ Post\ Treatment} = 0.740$; $p = 0.131$; two-tailed t test; $t(151) = 1.520$ ; $d_z = 0.12$). However, participants had better attitudes towards the outgroup after exposure to the video ( $M_{Index\ Outgroup\ Pre\ Treatment} = 0.242$; $M_{Index\ Outgroup\ Post\ Treatment} = 0.284$; $p < 0.001$; two-tailed t test; $t(151) = 5.583$ ; $d_z = 0.45$). The difference between the indices before and after exposure to the video of 0.042 index points corresponded to a difference of 12.6 points on the trust or openness to friendship scale from 0 to 100 or to a difference of 0.5 points on the fusion scale from 1 to 5. While this effect size was not particularly large in absolute terms and was definitely not equivalent to a change from potentially hostile to benevolent or friendly, the effect was highly significant and constituted at least some improvement with regards to the attitudes towards the outgroup. Given the high emotional intensity of the conflict, a single speech – regardless how powerful it is – might have its limits in improving attitudes towards the outgroup in any case.

Individual level absolute differences between attitudes towards Israelis and attitudes towards Palestinians decreased after exposure to the speech ( $M_{Index\ Difference\ Pre\ Treatment} = 0.468$; $M_{Index\ Difference\ Post\ Treatment} = 0.441$; $p < 0.001$; one-tailed t test; $t(158) = -3.9839$ ; $d_z = -0.32$; H2).

We also analysed heterogeneous effects in an exploratory manner. Focusing on attitudes towards the outgroup we found significant effects both for Jewish participants ($M_{Index\ Outgroup\ Pre\ Treatment}$ = 0.274; $M_{Index\ Outgroup\ Post\ Treatment}$ = 0.315; p < 0.001; two-tailed t test; t(106)= 5.0143 ; $d_z = 0.48$) and Muslim participants ($M_{Index\ Outgroup\ Pre\ Treatment}$ = 0.167; $M_{Index\ Outgroup\ Post\ Treatment}$ = 0.210; p = 0.011; two-tailed t test; t(44)= 2.6482 ; $d_z = 0.39$). The changes of the outgroup index for both groups were of similar size and we found no evidence of diverging effects ($M_{Effect\ Jewish\ participants}$ = 0.042; $M_{Effect\ Muslim\ participants}$ = 0.043 p = 0.955; two-tailed t test; t(150)= 0.0570 ; $d_z = 0.01$). We also ran an OLS regression with robust standard errors, the change of the outgroup index as the outcome variable and a religion dummy, age, a university education dummy, a female dummy, political orientation, psychological impact of news reporting, and index differences before exposure to the speech as independent variables and found no significant effects for neither variable (see table S3 in Appendix S3 in S1 File).

Moreover, we found that the effect of exposure to the speech on the outgroup index was significant for strongly polarized participants whose absolute difference between the Israeli index and the Palestinian index was larger than 0.5 index points before exposure to the speech (n = 71) which is a substantial difference since both indices range from 0 to 1 ($M_{Index\ Outgroup\ Pre\ Treatment}$ = 0.073; $M_{Index\ Outgroup\ Post\ Treatment}$ = 0.108; p = 0.004; two-tailed t test; t(70)= 2.9867 ; $d_z = 0.35$). This result suggests that the speech effectively improves attitudes towards the outgroup even for very strongly polarized participants.

With regard to the single components of the indices (i.e. identity fusion, trust and openness to friendship), we found highly significant correlations between all three variables (see tables S1 and S2 in Appendix S3 in S1 File). We also found significant treatment effects on identity fusion, trust and openness to friendship towards the outgroup (see Appendix S3 in S1 File).

## Study 2

In our second study, we used the same design as in the first study and recruited a general population sample of US citizens to analyse whether the effects we found in the first study also appear in the general population. We pre-registered the same hypotheses as for the first study, see AsPredicted #155,463: https://aspredicted.org/32X_X8Z

We recruited 400 participants on Prolific who were US citizens and neither Jews nor Muslims. All participants took our study on December 15, 2023. After excluding participants according to the same pre-registered criteria as in study 1, we had 361 participants left. We paid our participants £1.05 which corresponded to an average hourly wage of £8.24.

Participants were on average 44 years old, 59% had a university degree, 52% were females, and the average position on a political spectrum scale from 1 (left) to 10 (right) was 4.5. Fifty-five% of participants were Christians, 39% were atheists, and the remaining participants reported other religious beliefs. Forty-seven% of participants had better attitudes towards the Israeli side before treatment and 28% had better attitudes to the Palestinian side before treatment while 25% had perfectly identical index values before treatment. Participants reported an average psychological impact of news reporting on violence between Israelis and Palestinians (before exposure to treatment) of 2.36 which ranged between 2 (Little impact) and 3 (Medium impact) on a scale from 1 (No impact) to 5 (Extreme impact).

Participants had better attitudes towards Israelis after exposure to the video ($M_{Index\ Israelis\ Pre\ Treatment}$ = 0.451; $M_{Index\ Israelis\ Post\ Treatment}$ = 0.470; p < 0.001; one-tailed t test; t(360)=4.545 ; $d_z = 0.24$; H1). Participants also had better attitudes towards Palestinians after exposure to the video ($M_{Index\ Palestinians\ Pre\ Treatment}$ = 0.393; $M_{Index\ Palestinians\ Post\ Treatment}$ = 0.430; p < 0.001; one-tailed t test; t(360)=7.510 ; $d_z = 0.40$; H1).

Next, we look at participants who had identical and non-identical index values before treatment separately in an exploratory manner. The 25% of participants who had identical index values before treatment had better attitudes towards Israelis ($M_{Index\ Israelis\ Pre\ Treatment}$ = 0.439; $M_{Index\ Israelis\ Post\ Treatment}$ = 0.464; p = 0.002; two-tailed t test; t(89)=3.225 ; $d_z = 0.34$) and Palestinians ($M_{Index\ Palestinians\ Pre\ Treatment}$ = 0.439; $M_{Index\ Palestinians\ Post\ Treatment}$ = 0.465; p = 0.001; two-tailed t test; t(89)=3.304 ; $d_z = 0.35$) after exposure to the video, showing that the treatment could also affect those who were neutral.

Those who preferred one side over the other before treatment had better attitudes towards their ingroup after treatment ($M_{Index\ Ingroup\ Pre\ Treatment}$= 0.512; $M_{Index\ Ingroup\ Post\ Treatment}$=0.522; p=0.049; two-tailed t test; t(270)=1.980 ; $d_z = 0.12$) and towards their outgroup after treatment ($M_{Index\ Outgroup\ Pre\ Treatment}$= 0.320; $M_{Index\ Outgroup\ Post\ Treatment}$=0.369; p<0.001; two-tailed t test; t(270)=8.350 ; $d_z = 0.51$). The change in the ingroup index was larger than the change in the outgroup index ($M_{Change\ Ingroup\ Index}$= 0.010; $M_{Change\ Outgroup\ Index}$=0.049; p<0.001; two-tailed t test; t(270)=6.365 ; $d_z = 0.39$).

Individual level absolute differences between attitudes towards Israelis and attitudes towards Palestinians decreased ($M_{Index\ Difference\ Pre\ Treatment}$= 0.144; $M_{Index\ Difference\ Post\ Treatment}$=0.126; p<0.001; one-tailed t test; t(360)=−4.055 ; $d_z = -0.21$; H2).

We also analysed heterogeneous effects in an exploratory manner. We ran an OLS regression with robust standard errors, the change of the outgroup index as the outcome variable and age, a university education dummy, a female dummy, political orientation, psychological impact of news reporting, and index differences before exposure to the speech as independent variables and found no significant effects for neither variable (see table S6 in Appendix S4 in S1 File).

We also found that the treatment effect on the outgroup index was significant for polarized participants whose absolute difference between the Israeli index and the Palestinian index was larger than 0.2 index points before exposure to thespeech (n=91 which is 25% if the sample) which is a notable difference since both indices range from 0 to 1 ($M_{Index\ Outgroup\ Pre\ Treatment}$= 0.200; $M_{Index\ Outgroup\ Post\ Treatment}$=0.252; p<0.001; two-tailed t test; t(90)= −4.6232 ; $d_z = 0.48$). This result shows that the speech effectively improved attitudes towards the outgroup even among polarized participants.

With regard to the single components of the indices (i.e. identity fusion, trust and openness to friendship), we found highly significant correlations between all three variables (see tables S4 and S5 in Appendix S4 in S1 File). We also found significant effects on identity fusion, trust and openness to friendship towards the outgroup (see section Appendix S4 in S1 File).

## Wave 2

We re-contacted the participants who were included in the main analysis three days after the respective first wave started and invited them to participate in a second wave. We wanted to investigate whether the effects fully or partially remained. See AsPredicted #155,542: https://aspredicted.org/4CT_H1F. We started wave 2 for study 1 on December 17, 2023 and ended it as pre-registered on December 20, 2023. Out of the 159 participants of study 1, 149 also took wave 2. We paid these participants £0.30 for wave 2 which corresponded to an average hourly wage of £16.62. The median difference between the points in time where participants took both waves was 71 hours. In the following, all results concerning study 1 will include only participants who also took wave 2. In the following we report results for attitudes towards ingroup and outgroup rather than towards Israelis and Palestinians as we had pre-registered since we think that looking at attitudes towards ingroup and outgroup is more informative. We report the analyses that focus on the Israeli index and the Palestinian index in Appendix S5 in S1 File.

Although there was no significant effect on attitudes towards the ingroup in study 1 ($M_{Index\ Ingroup\ Pre\ Treatment}$= 0.730; $M_{Index\ Ingroup\ Post\ Treatment}$=0.738; p=0.134; two-tailed t test; t(143)=1.507 ; $d_z =0.13$) participants had worse attitudes towards the ingroup in wave 2 ($M_{Index\ Ingroup\ Pre\ Treatment}$= 0.730; $M_{Index\ Ingroup\ Wave\ 2}$=0.705; p<0.001; two-tailed t test; t(143)=−3.479 ; $d_z = -0.29$). Perhaps this result was due to critical reflection on the ingroup that was caused by the video. Participants in study 1 felt very strongly about their ingroup and hence there was certainly some potential for critical reflection. Another possibility is a time trend, although we are not aware that something unusual happened between both waves. We also did not observe such a decrease for the participants in study 2 who felt less strongly about the group they favoured (see below) which limits the potential for critical reflection to meaningfully change attitudes. The effect on attitudes towards the outgroup ($M_{Index\ Outgroup\ Pre\ Treatment}$= 0.239; $M_{Index\ Outgroup\ Post\ Treatment}$=0.280; p<0.001; two-tailed t test; t(143)=5.337 ; $d_z = 0.44$) partially persisted ($M_{Index\ Outgroup\ Pre\ Treatment}$= 0.239; $M_{Index\ Outgroup\ Wave\ 2}$=0.261; p=0.002; two-tailed t test; t(143)=3.141 ; $d_z = 0.26$).

The effect on individual level absolute differences between attitudes towards Israelis and attitudes towards Palestinians ( $M_{Index\ Difference\ Pre\ Treatment}$ = 0.475; $M_{Index\ Difference\ Post\ Treatment}$ = 0.447; p < 0.001; one-tailed t test; t(148)=−3.989 ; $d_z$ = −0.33) persisted fully ( $M_{Index\ Difference\ Pre\ Treatment}$ = 0.475; $M_{Index\ Difference\ Wave\ 2}$ = 0.436; p < 0.001; one-tailed t test; t(148)=−4.3421 ; $d_z$ = –0.63; H2).

Out of the 361 participants of study 2, 338 also took wave 2. We started wave 2 for study 2 on December 18, 2023 and ended it as pre-registered on December 21, 2023. We paid these participants £0.30 for the wave 2 which corresponded to an average hourly wage of £16.62. The median difference between the points in time where participants took both waves was 72 hours. In the following, all results concerning study 2 will include only participants who also took wave 2.

In study 2, the effect on attitudes towards the ingroup ( $M_{Index\ Ingroup\ Pre\ Treatment}$ = 0.510; $M_{Index\ Ingroup\ Post\ Treatment}$ = 0.523; p = 0.008; two-tailed t test; t(253)=2.659 ; $d_z$ = 0.17) vanished completely ( $M_{Index\ Ingroup\ Pre\ Treatment}$ = 0.510; $M_{Index\ Ingroup\ Wave\ 2}$ = 0.512; p = 0.725; two-tailed t test; t(253)=0.352 ; $d_z$ = 0.02). The effect on attitudes towards the outgroup ( $M_{Index\ Outgroup\ Pre\ Treatment}$ = 0.322; $M_{Index\ Outgroup\ Post\ Treatment}$ = 0.372; p < 0.001; two-tailed t test; t(253)=8.275 ; $d_z$ = 0.52) persisted partially ( $M_{Index\ Outgroup\ Pre\ Treatment}$ = 0.322; $M_{Index\ Outgroup\ Wave\ 2}$ = 0.351; p < 0.001; two-tailed t test; t(253)=4.062 ; $d_z$ = 0.25).

The effect on individual level differences between attitudes towards Israelis and attitudes towards Palestinians ( $M_{Index\ Difference\ Pre\ Treatment}$ = 0.142; $M_{Index\ Difference\ Post\ Treatment}$ = 0.125; p < 0.001; one-tailed t test; t(337)=−3.981 ; $d_z$ = –0.22) was no longer significant ( $M_{Index\ Difference\ Pre\ Treatment}$ = 0.142; $M_{Index\ Difference\ Wave\ 2}$ = 0.136; p = 0.160; one-tailed t test; t(337)=−0.997 ; $d_z$ = –0.05; H2). However, when restricting the sample in an exploratory manner to the half of the participants who were the most polarized before treatment (i.e., those with a higher absolute pre-treatment difference than the median of 0.05 index points), then the effect ( $M_{Index\ Difference\ Pre\ Treatment}$ = 0.283; $M_{Index\ Difference\ Post\ Treatment}$ = 0.248; p < 0.001; two-tailed t test; t(163)=4.449 ; $d_z$ = –0.35) was persistent ( $M_{Index\ Difference\ Pre\ Treatment}$ = 0.283; $M_{Index\ Difference\ Wave\ 2}$ = 0.256; p = 0.006; two-tailed t test; t(163)=−2.797 ; $d_z$ = –0.22). Participants with a pre-treatment difference of 0.05 index points or lower were not strongly polarized anyway and since the video was unlikely to make them feel more partisan, they might just have added noise to the long-term effect on index differences.

## Discussion

We have shown that exposure to a speech appealing to shared suffering in the Israeli-Palestinian conflict improved attitudes towards the outgroup for US citizens identifying as Muslim or Jewish as well as for a general population sample from the US. We also showed that these positive effects were partially but significantly persistent after three days. These results suggest that appealing to shared suffering may be an effective strategy to mitigate tension even in the heated climate of Israeli-Palestinian relations in December 2023. The speech we used as a treatment was delivered by a Western politician who has actively participated in the real-world debate about the Israeli-Palestinian conflict which gave our treatment high real-world relevance. In exploratory analyses we found no evidence for heterogeneous effects with regard to age, education, gender, political spectrum, psychological impact of news reporting about the conflict, and pre-treatment index differences. We found significant effects of similar size for Muslims and Jews in study 1 and significant effects even for participants who were highly polarized before exposure to the speech in both studies. These results suggest that the speech is effective across a broad range of audiences.

We contribute to the identity fusion literature by demonstrating that fusion to the outgroup can be strengthened by appeals to shared dysphoric experiences shared by both groups. Fusion theory proposes that threats to the ingroup are taken personally by fused individuals [3,41,42], motivating extreme forms of pro-group action including willingness to fight and die to protect the ingroup. In line with these previous findings, appeals to shared suffering of ingroup *and* outgroup could also plausibly exacerbate outgroup hostility, but our results show that this is not necessarily the case – even in the very heated climate in December 2023 when the data collection took place. We also contribute to a literature on the

transformation of competitive victimhood beliefs into inclusive victimhood beliefs in the Israeli-Palestinian conflict [21,31] by showing that speeches about shared suffering can facilitate such a transformation.

Furthermore, we contribute to a literature on the reduction of animosity towards a political outgroup that analyses a rich toolbox of interventions including correcting misperceptions about the outgroup [43], emphasizing common interests [44], highlighting commonalities [45], creating real shared experiences [46], fostering intergroup contact [47], and modelling warm intergroup relations [48]. The intervention we analysed – appeals to shared suffering in a speech – was effective even when intergroup animosity was widespread. This intervention is easily scalable (ranging from private conversations to public speeches by political leaders) and does not require a lot of resources or the cooperation of the outgroup.

An obvious limitation of our study was that we only had US samples. However, US citizens of Muslim and Jewish faith were emotionally involved in the conflict and were strongly polarized. Moreover, the psychological links between shared dysphoric experiences, intense group bonding, and extreme forms of pro-group action have been studied in diverse populations including agro-pastoral combatants in rural Cameroon [49], Muslim fundamentalists in Indonesia [50], Northern Irish Republicans and Unionists [9], Brazilian football hooligans [51], and Libyan insurgents [11]. However, future research might test whether similar interventions also improve attitudes towards the outgroup in Israeli and Palestinian samples and might also look at heterogeneous effects with regard to the degree of personal loss suffered in the conflict. Future research might also analyse the impact of similar interventions in other conflicts and might possibly investigate whether characteristics of the speaker affect the impact of the speech.

A second limitation of our studies was that we did not have a between-subjects design with multiple conditions. Other treatment conditions may have varied the speeches to explore how distinct elements of the speech (e.g., the personality of Lord Alderdice, or the concrete examples for shared suffering the speech gave) impact various attitudes towards the outgroup. A control condition would also have allowed us to exclude time trends as an explanation for the effects we found in the second wave. However, we are not aware of any unusual confounding events or changes in the political climate in the time span between the waves that could explain the effects. We decided to use the within-subjects design due to the limited number of available US participants of Muslim and Jewish faith, but future research using between-subjects designs is certainly desirable.

A third limitation of our study was that we only have data on outcomes three days after exposure to the speech. Future research on the durability of interventions after longer time periods is certainly desirable. Nevertheless, our study took place in December 2023 and participants – especially Muslim and Jewish ones - were likely exposed to other intense real-world stimuli related to the conflict between both waves such as social media content, news reports about violence in the conflict or emotional personal discussions. Also, many of the participants who were strongly polarized in our study were likely exposed to real-world stimuli that primarily made the suffering of their ingroup salient. In this light, the partially persistent effects of one single 3-minute speech in video format seem to demonstrate the effectiveness of our approach. This is consistent with identity fusion theory since the perceptions of shared essence that are caused by the shared experiences pathway to fusion are not merely a fleeting state of mind that easily comes and goes but rather an enduring psychological trait, once established. However, as a single exposure to a 3-minute speech in video format is most likely not enough to fully counter the effects of decades of conflict and trauma, future research might also analyse the long-term impacts of repeated exposure to interventions that highlight shared suffering across group boundaries as well as interventions that highlight shared suffering in a more direct and intense way, e.g. through personal contact with an outgroup member who has directly suffered in the conflict.

A potential concern for the interpretation of our results are experimenter demand effects [52,53]. Participants may have believed that the study was intended to show that the speech would improve attitudes towards the outgroup. Alternatively, participants may have believed that the study was intended to trigger angry reactions. Nevertheless, based on replications of five experimental designs touching political topics, Mummolo and Peterson [54] found that providing information about

the experimenter's intent did not alter treatment effects in their online survey experiments. Bursztyn and colleagues [55] present evidence that the intrinsic willingness to answer anonymous questions in line with one's own political convictions can be quite strong in the context of anti-American attitudes of Pakistani participants even in the presence of substantial monetary incentives against stating these attitudes.

Investigating socially loaded topics can also induce social desirability bias. Participants might have felt that the socially desirable reaction to the speech would be to express favourable attitudes towards both sides. However, the opposite response would also have been plausible. For example, those who reported very favourable attitudes towards the ingroup and very unfavourable attitudes towards the outgroup may have considered that among their peers the socially desirable reaction to references to suffering of the outgroup or suffering on both sides would be to double down on a partisan stance. In any case, it is not necessarily the case that political speeches generate an effect that goes into the direction of the arguments the speech championed. For instance, Wuttke and colleagues [35] find null effects of pro-democracy speeches delivered by prominent Republican politicians in the aftermath of January sixth on commitment to democratic norms among Republican voters.

Lasting peace in the Israeli-Palestinian conflict requires political solutions, but political solutions depend on establishing the right psychological foundations which are often missing from conflict resolution efforts. The suffering of the other side in many conflicts is widely overlooked in practice and even actively concealed by propaganda on both sides. Moreover, the rising use of social media as a propaganda instrument can contribute to polarization and us-vs-them thinking, exacerbating the need for efforts to reduce tensions. Our work suggests that greater recognition of shared sufferings in political speeches, news reporting, and other forms of public discourse might help to meet that need.

## Supporting information

**S1 File. Supplementary Material: We have uploaded an Appendix as supplementary materials.**
(PDF)

**S1 Data. Data study 1 Stata format.**
(DTA)

**S2 File. Data study 1 Excel format.**
(XLSX)

**S2 Data. Data study 2 Stata format.**
(DTA)

**S3 File. Data study 2 Excel format.**
(XLSX)

**S3 Data. Data study 1 both Waves Stata format.**
(DTA)

**S4 File. Data study 1 both Waves Excel format.**
(XLSX)

**S4 Data. Data study 2 both Waves Stata format.**
(DTA)

**S5 File. Data study 2 both Waves Excel format.**
(XLSX)

## Acknowledgments

We thank Charlotte Hamm and Joris Lammers for helpful comments. We also thank our reviewers Assaf Suberry, Eyal Lewin, and an anonymous reviewer for valuable and constructive feedback.

## Author contributions

**Conceptualization:** Lukas Reinhardt, Harvey Whitehouse.

**Data curation:** Lukas Reinhardt, Harvey Whitehouse.

**Formal analysis:** Lukas Reinhardt.

**Funding acquisition:** Harvey Whitehouse.

**Writing – original draft:** Lukas Reinhardt, Harvey Whitehouse.

**Writing – review & editing:** Lukas Reinhardt, Harvey Whitehouse.

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
