## [Decision Letter · Decision Letter 0]

12 Mar 2025

PONE-D-25-02752Appeals to Shared Suffering in the Context of the Israeli-Palestinian ConflictPLOS ONE

Dear Dr. Reinhardt,

Thank you for submitting your manuscript to PLOS ONE. After careful consideration, we feel that it has merit but does not fully meet PLOS ONE’s publication criteria as it currently stands. Therefore, we invite you to submit a revised version of the manuscript that addresses the points raised during the review process.

Dear author

Thank you for submitting your paper to to PLOS One.

Following the comments of the two reviewers, I am returning the article to you for revisions.

Please note all the comments and address them.

Mark the relevant changes in the body of the article. We will wait to your resubmission after the revisions.

Best regards

We look forward to receiving your revised manuscript.

Kind regards,

Gal Harpaz, Ph.D.

Academic Editor

PLOS ONE

Journal Requirements:

3. Please remove all personal information, ensure that the data shared are in accordance with participant consent, and re-upload a fully anonymized data set. 

Reviewers' comments:

Reviewer's Responses to Questions

**Comments to the Author**

1. Is the manuscript technically sound, and do the data support the conclusions?

Reviewer #1: Yes

Reviewer #2: Yes

2. Has the statistical analysis been performed appropriately and rigorously? 

Reviewer #1: Yes

Reviewer #2: Yes

3. Have the authors made all data underlying the findings in their manuscript fully available?

Reviewer #1: No

Reviewer #2: Yes

4. Is the manuscript presented in an intelligible fashion and written in standard English?

Reviewer #1: Yes

Reviewer #2: Yes

5. Review Comments to the Author

Reviewer #1: This research delves into the significant topic of how acknowledging shared suffering in the Israeli-Palestinian conflict can improve intergroup attitudes. By presenting evidence from studies conducted with both Muslim and Jewish Americans and a broader US population, it demonstrates that appeals to shared suffering can foster positive attitudes towards outgroups, with effects persisting even days after exposure. I appreciate the opportunity to review this important topic and acknowledge the valuable work of the authors. Here, I will outline some of the paper's accomplishments and also highlight concerns that should be addressed to enhance the manuscript.

Introduction: The literature review of the introduction section represents a good and sound background for the study.

I appreciate that the authors have explicitly acknowledged the wording choices, Lord John Alderdice’s personality. By addressing these factors, they demonstrate an awareness of potential limitations. This transparency strengthens the study's credibility.

Major:

Abstract: please add a description of the research sample and main results (e.g., p – values).

Page 4, line 10: Authors declare: “we chose a within-subjects design” can you elaborate more about this restriction? How many participants are needed for between-subject design? As such an analysis can provide information about the different impact of the intervention (i.e., the speech) across the two ethnic groups.

Page 4, 12-15 - I think it will be valuable to provide a more detailed analysis of the scale that was consolidated from three distinct measures or facets (namely, identity fusion, trust and openness to friendship). The measure was called by authors as “attitudes” but actually refer to different things. Did the authors conduct a statistical analysis for each facet? It would be interesting to know which one was mostly affected by the intervention.

Page 6, line 23-25: I am unclear about the relevance of mentioning the number of participants for each of the three days. Was there distinct or significant news about the war during these days? Did you examine the variations across them? It appears to me that you did not. Therefore, I recommend only mentioning the total sample after the three days.

Results: Page 6. I believe providing a demographic description and an analysis of the differences between the groups (American Muslim vs. American Jewish participants), particularly concerning age and gender, would be valuable. If significant differences between the groups are identified, you should consider controlling for them in subsequent analyses.

Page 7: The choice between a one-tailed or two-tailed test hinges on whether the research hypothesis predicts a specific direction for the effect, which necessitates a one-tailed test, or merely indicates that there is a difference, requiring a two-tailed test. is there a particular reason for selecting the two-tailed t-test instead of the one-tailed t-test?

Additionally, the t-test results do not completely report the t-value.

The authors clearly understand the intense emotional conflict involved. I wonder if they have considered measuring the participants' emotional involvement, such as experiencing emotional distress or having family or friends injured in the conflict. These variables could add unique insights to the characteristics of this population.

Discussion: Although the introduction is well-structured, the discussion requires further elaboration. How does this work contribute to existing literature and theoretical understanding? Additionally, consider how other communication tools might help in resolving such conflicts, perhaps by facilitating dialogue, promoting understanding, or easing tensions between parties.

It would be valuable to elaborate on the implications of the research on the real world conflict. What the authors think the effect of exposure to such speech on Israelis (directly injured vs. not directly injured in both side, e.g., Israelis that live in the center of Israel vs Gaza Envelope / Palestinians living in their permanent house vs those that were evacuated due to the IDF invasion to Gaza). How other conflicts can benefit from the research (e.g., the Ukraine – Russia war, animosity between ethnic groups in Africa esc.)

Minor:

Page 2, Line 39: consider deleting the words “the power of” in order to keep the tone descriptive.

The additional files attached to the manuscript, such as .do or .dta files, are not accessible. Is there an alternative way to approach them? Additionally, the PDF supplementary file is well structured, consisting of seven sections. I recommend labeling each section as Appendix 1, Appendix 2, etc. Furthermore, while the manuscript refers readers to Appendices 1, 2, and 6, please consider directing them to the other important information available in Appendices 3, 4, 5, and 7.

Reviewer #2: The paper presents a noteworthy and innovative research concept, with an original experimental design. The effort to identify real-world mechanisms for reducing intergroup hostility is particularly commendable. Overall, the article appears well-suited for publication.

However, I have some reservations regarding the selection of Lord John Alderdice as the speaker. Firstly, the paper does not provide a detailed account of the selection process, including the criteria used, alternative candidates considered, and the specific rationale for choosing Alderdice as the optimal speaker. Secondly, the significance of introducing an Irish figure to an American audience warrants further discussion. Would it not be more effective to present a speaker whose background and perspectives might resonate more closely with the local audience? Thirdly, given Alderdice’s selection, the paper would benefit from a more comprehensive discussion of his biography and professional standing. Beyond serving as a justification for his inclusion, such information contributes to the transparency and accountability of the research. Finally, while the inclusion of a video link to the speech is valuable, it would be advisable to also provide a written transcript of the speech for accessibility and reference.

6. PLOS authors have the option to publish the peer review history of their article (what does this mean? ). If published, this will include your full peer review and any attached files.

**Do you want your identity to be public for this peer review?** For information about this choice, including consent withdrawal, please see our Privacy Policy .

Reviewer #1: **Yes: ** Dr. Assaf Suberry

Reviewer #2: **Yes: ** Dr. Eyal Lewin - Ariel Univsersity

---

## [Author Response · Author response to Decision Letter 1]

2 May 2025

Response to reviewers

We thank both reviewers for their time and their constructive and valuable feedback. We respond to their comments below.

Comments from Reviewer #1

Major:

Comment 1: Abstract: please add a description of the research sample and main results (e.g., p – values).

Response: We have now added information about the sample sizes of both studies in the abstract to complement our descriptions of the samples (i.e. ‘Muslim and Jewish Americans’ in the first study and ‘sample of the general US population’ in the second study). We have also noted that exposure to the speech fostered identity fusion, trust and openness to friendship towards the outgroup rather than just referring to attitudes. Since our results are highly significant and it’s somewhat unusual in our fields to state p-values in the abstract we would prefer to refrain from doing so unless this is a strict requirement.

Comment 2: Page 4, line 10: Authors declare: “we chose a within-subjects design” can you elaborate more about this restriction? How many participants are needed for between-subject design? As such an analysis can provide information about the different impact of the intervention (i.e., the speech) across the two ethnic groups.

Response: We now explain in the section ‘Introduction/Present research’ that a within-subjects design increases power because it creates two rather than one data point per participant and controls for individual-specific effects by design (Charness et al., 2012). In order to obtain the same power, between-subjects designs can require four times as many participants as within-subjects designs, although this ratio can be higher or lower depending on the relevant parameters (List, 2025). We also observe that the within-subjects design allows us to obtain treatment effects on an individual level which facilitates the analysis of heterogeneous treatment effects even in relatively small samples.

List, J. A. (2025). The Experimentalist Looks Within: Toward an Understanding of Within-Subject Experimental Designs (No. w33456). National Bureau of Economic Research Working Paper Series.

Charness, G., Gneezy, U., & Kuhn, M. A. (2012). Experimental methods: Between-subject and within-subject design. Journal of economic behavior & organization, 81(1), 1-8.

We also added exploratory analyses of heterogeneous effects in the sections ‘Study 1/Results’ and ‘Study 2’. In both studies, we found no evidence of heterogeneous effects with regard to age, education, gender, position on the political spectrum, psychological impact of news reporting about the conflict, and pre-treatment index differences. We found significant effects of similar size for Muslims and Jews in study 1 and significant effects even for participants who were highly polarized before exposure to the speech in both studies. These results suggest that the speech is effective across a broad range of audiences.

Comment 3: Page 4, 12-15 - I think it will be valuable to provide a more detailed analysis of the scale that was consolidated from three distinct measures or facets (namely, identity fusion, trust and openness to friendship). The measure was called by authors as “attitudes” but actually refer to different things. Did the authors conduct a statistical analysis for each facet? It would be interesting to know which one was mostly affected by the intervention.

Response: We agree that the effects on three single outcome variables are interesting. We present correlations between these three variables in tables S1, S2, S4 and S5 in the Appendix and observe highly significant and strong correlations. We also present effects on the single outcome variables in study 1 and study 2 in Appendix S3 and Appendix S4 and we found significant effects for all three variables. We have now added remarks about these results in the main manuscript at the end of section ‘Study 1/Results’ and section ‘Study 2’.

Comment 4: Page 6, line 23-25: I am unclear about the relevance of mentioning the number of participants for each of the three days. Was there distinct or significant news about the war during these days? Did you examine the variations across them? It appears to me that you did not. Therefore, I recommend only mentioning the total sample after the three days.

Response: We indicated the number of participants who took part on each date to illustrate that we stopped data collection after day 3 because we did not expect many further participants to join after this day. We explain this before we turn to describing the sample characteristics in section ‘Study 1/Data’.

Comment 5: Results: Page 6. I believe providing a demographic description and an analysis of the differences between the groups (American Muslim vs. American Jewish participants), particularly concerning age and gender, would be valuable. If significant differences between the groups are identified, you should consider controlling for them in subsequent analyses.

Response: We have now added further information about differences between Jewish and Muslim participants at the end of section ‘Study 1/Data’. Jewish participants were significantly older, more left-wing, and reported less psychological impact of news reporting on the violence between Israelis and Palestinians. There were no significant differences regarding the shares of university educated participants or female participants. However, we also now emphasize that comparisons between Jewish and Muslim participants are not our focus. Moreover, in the one comparative analysis we added we found very similar effects of exposure to the speech on the outgroup index for both groups (Effect of speech on outgroup index among Jewish participants: 0.042; Effect of speech on outgroup index among Muslim participants: 0.043; p-value of the t-test comparing effect sizes between both groups: p=0.955). See also our response to your second comment where we explain that we did not find evidence for heterogeneous treatment effects with regard to demographic characteristics.

Comment 6: Page 7: The choice between a one-tailed or two-tailed test hinges on whether the research hypothesis predicts a specific direction for the effect, which necessitates a one-tailed test, or merely indicates that there is a difference, requiring a two-tailed test. is there a particular reason for selecting the two-tailed t-test instead of the one-tailed t-test?

Response: This point is well taken, and we now report one-tailed t-tests for analyses that test our directional hypotheses.

Comment 7: Additionally, the t-test results do not completely report the t-value.

Response: We now also report t-values.

Comment 8: The authors clearly understand the intense emotional conflict involved. I wonder if they have considered measuring the participants' emotional involvement, such as experiencing emotional distress or having family or friends injured in the conflict. These variables could add unique insights to the characteristics of this population.

Response: We included a psychological impact measure in the demographics block of the survey before participants were exposed to the speech. The question was ‘To what extent has news reporting on violence between Israelis and Palestinians impacted you psychologically?’ with answers ‘No impact’ (1), ‘Little impact’ (2), ‘Medium impact’ (3), ‘Strong impact’ (4), and ‘Extreme impact’ (5). Participants in study 1 (Jews and Muslims) reported an average impact of 3.57 and participants from the general US population (study 2) reported an average impact of 2.36. We have added now remarks on this in the second paragraph of section ‘Study 1/Data’ and in the third paragraph of section ‘Study 2’.

Comment 9: Discussion: Although the introduction is well-structured, the discussion requires further elaboration. How does this work contribute to existing literature and theoretical understanding? Additionally, consider how other communication tools might help in resolving such conflicts, perhaps by facilitating dialogue, promoting understanding, or easing tensions between parties.

Response: We elaborate further on our contribution to the literature in the Discussion. First, we discuss our contribution to the identity fusion literature. According to identity fusion theory, threats to the ingroup are taken personally by fused participants motivating extreme forms of pro-group action to protect the ingroup from threat which can motivate outgroup hostility. We contribute to the fusion literature by showing that highlighting suffering of ingroup and outgroup simultaneously does not necessarily lead to exacerbated intergroup conflict but can also help to heal divisions. We also elaborate that we contribute to the literature on transforming competitive victim beliefs into inclusive victim beliefs in the context of the Israeli-Palestinian conflict by demonstrating that speeches can effectively facilitate such a transformation. Finally, we situate our work within a literature on interventions that reduce animosity towards a political outgroup and discuss other tools that may also be designed for this purpose including (among others) correcting misperceptions about the outgroup, highlighting common interests, and highlighting commonalities.

Comment 10: It would be valuable to elaborate on the implications of the research on the real world conflict. What the authors think the effect of exposure to such speech on Israelis (directly injured vs. not directly injured in both side, e.g., Israelis that live in the center of Israel vs Gaza Envelope / Palestinians living in their permanent house vs those that were evacuated due to the IDF invasion to Gaza). How other conflicts can benefit from the research (e.g., the Ukraine – Russia war, animosity between ethnic groups in Africa esc.)

Response: We have now added some remarks to address this point in the fourth paragraph the Discussion after our discussion of the limitation that we only include US participants. The cross-cultural ubiquity of the link between perceptions of shared dysphoric experiences and intense group bonding as well as the fact that we find effects even for strongly polarized participants in our samples can be seen as arguments that similar interventions might also work in Israeli and Palestinian samples and in the context of other conflicts. However, we also note that future research might test the impact of similar interventions in these samples. We also note that future research might look at heterogeneous effects with regard to the degree of direct personal loss suffered in the conflict.

Minor:

Comment 11: Page 2, Line 39: consider deleting the words “the power of” in order to keep the tone descriptive.

Response: We have adopted this suggestion.

Comment 12: The additional files attached to the manuscript, such as .do or .dta files, are not accessible. Is there an alternative way to approach them? Additionally, the PDF supplementary file is well structured, consisting of seven sections. I recommend labeling each section as Appendix 1, Appendix 2, etc. Furthermore, while the manuscript refers readers to Appendices 1, 2, and 6, please consider directing them to the other important information available in Appendices 3, 4, 5, and 7.

Response: We will re-upload these files as part of our resubmission in case there were any technical issues with the files previously uploaded. Reviewer 2 indicated that the data was available to him. That said, we used Stata for the analyses and the .do file can also be opened with a text-editor and the .dta files can be imported and converted into other formats with other data analysis programmes. As recommended, we have also re-labelled the Appendix sections, and we are now referring to all Appendices in the main text. As recommended by reviewer 2, we have moved the transcript into the main manuscript, thus we have now 6 sections in the Appendix rather than 7.

Comments from Reviewer #2

Comment 1: The paper presents a noteworthy and innovative research concept, with an original experimental design. The effort to identify real-world mechanisms for reducing intergroup hostility is particularly commendable. Overall, the article appears well-suited for publication.

However, I have some reservations regarding the selection of Lord John Alderdice as the speaker. Firstly, the paper does not provide a detailed account of the selection process, including the criteria used, alternative candidates considered, and the specific rationale for choosing Alderdice as the optimal speaker. Secondly, the significance of introducing an Irish figure to an American audience warrants further discussion. Would it not be more effective to present a speaker whose background and perspectives might resonate more closely with the local audience? Thirdly, given Alderdice’s selection, the paper would benefit from a more comprehensive discussion of his biography and professional standing. Beyond serving as a justification for his inclusion, such information contributes to the transparency and accountability of the research. Finally, while the inclusion of a video link to the speech is valuable, it would be advisable to also provide a written transcript of the speech for accessibility and reference.

Response: We added a paragraph about Lord Alderdice’s background in section ‘Study 1/Design and hypotheses’. Lord Alderdice is a member of the House of Lords and the Select Committee of the House on International Relations and Defence. He also played a significant role in the Irish Peace Process as a former leader of the Alliance Party of Northern Ireland. We selected him as a speaker because of his substantial expertise on intractable conflicts. We acknowledge that he might be perceived differently than a US speaker by a US audience. However, we also note that this feature of our design allows us to rule out interaction effects with partisan affiliation or prior beliefs about the speaker. Thus, our setup might be in a position to investigate the impact of the speech per se without these additional layers. We have also added a remark in the fourth paragraph of the Discussion that future research might investigate whether characteristics of the speaker affect the impact of the speech, and we agree that the characteristics of the speaker are an interesting aspect of the study design. We have included a written transcript of the speech in section ‘Study 1/Transcript of the speech’ of the main manuscript.

---

## [Decision Letter · Decision Letter 1]

3 Jul 2025

PONE-D-25-02752R1Appeals to shared suffering in the context of the Israeli-Palestinian conflictPLOS ONE

Dear Dr. Reinhardt,

Thank you for submitting your manuscript to PLOS ONE. After careful consideration, we feel that it has merit but does not fully meet PLOS ONE’s publication criteria as it currently stands. Therefore, we invite you to submit a revised version of the manuscript that addresses the points raised during the review process.

Dear Author,

Thank you for resubmitting your manuscript to [Journal Name]. After careful consideration, I am pleased to inform you that we would be happy to consider a revised version pending minor revisions.

In particular, I encourage you to address the points raised by Reviewer 3, who made several constructive suggestions regarding some sections in your article.

Please submit a revised manuscript along with a brief response to the reviewer’s comments.

We look forward to receiving your revision.

Best regards,

We look forward to receiving your revised manuscript.

Kind regards,

Gal Harpaz, Ph.D.

Academic Editor

PLOS ONE

Journal Requirements:

Reviewers' comments:

Reviewer's Responses to Questions

**Comments to the Author**

1. If the authors have adequately addressed your comments raised in a previous round of review and you feel that this manuscript is now acceptable for publication, you may indicate that here to bypass the “Comments to the Author” section, enter your conflict of interest statement in the “Confidential to Editor” section, and submit your "Accept" recommendation.

Reviewer #1: All comments have been addressed

Reviewer #3: All comments have been addressed

2. Is the manuscript technically sound, and do the data support the conclusions?

Reviewer #1: Yes

Reviewer #3: Yes

3. Has the statistical analysis been performed appropriately and rigorously? 

Reviewer #1: Yes

Reviewer #3: Yes

4. Have the authors made all data underlying the findings in their manuscript fully available?

Reviewer #1: Yes

Reviewer #3: Yes

5. Is the manuscript presented in an intelligible fashion and written in standard English?

Reviewer #1: Yes

Reviewer #3: Yes

6. Review Comments to the Author

Reviewer #1: Dear Authors,

This is a timely and relevant topic for our world, where conflict resolution techniques and methods are of urgent and increasing importance. I appreciate the time and effort you have invested in revising the manuscript. The revised data analysis and discussion are much clearer, particularly the breakdown of “attitudes” into its three components. Based on these improvements, I believe the manuscript is now ready for publication.

Minor Comments:

On page 5, the authors state:

“The individual pictured in Fig 1 has provided written informed consent (as outlined in PLOS consent form) to publish their image alongside the manuscript.”

I recommend revising this for clarity and consistency:

“The individual pictured in Figure 1 has provided written informed consent (as outlined in the PLOSOne consent form) to publish his image alongside the manuscript.”

As noted in the first round of review, please consider including the main results in the abstract. You have added the sample size, which is helpful, but including the key outcomes and p-values would be more beneficial for readers who are screening the abstract of your manuscript.

Best regards

Reviewer #3: I believe that the manuscript presents an original and timely contribution on a topic of high contemporary relevance. It is evident that the authors have put considerable effort into addressing the reviewers’ comments from the previous round, and the revised version reflects substantial improvement. In my view, the manuscript is now close to being suitable for publication. I offer several comments, most of which are minor in nature:

Introduction – Background on the War

The opening paragraph, which provides a chronological overview of recent events in the Israel-Gaza war, could be significantly shortened. I am uncertain about the relevance of this historical background for the reader, who is likely already familiar with the context. Additionally, although the authors make a commendable effort to present the events in a neutral and objective tone, some readers who strongly identify with either the Israeli or Palestinian side may perceive the description as biased or unfair. Given the ongoing nature of the war and the emotional sensitivity of the topic, this paragraph risks alienating readers and undermining trust in the research. I recommend either substantially condensing the paragraph or removing it altogether, and beginning the introduction with the theoretical background—which, in my view, should be the core of the introduction.

Reporting Differences Across Outcome Dimensions

While I understand the rationale for creating a composite outcome index from identity fusion, trust, and openness to friendship (and I acknowledge the high intercorrelations reported in the appendix), the three variables represent conceptually distinct constructs. It would be valuable to examine whether the speech had different effects across these dimensions. If the effects were similar, the authors could briefly note this in the text and include the detailed results in the appendix. Either way, some comment in the main text would help clarify the distinctiveness (or lack thereof) of the outcome components.

Effect Size Reporting

The results section would benefit from more explicit reporting of effect sizes, such as Cohen’s d. While the authors describe mean changes and provide p-values, including standardized effect sizes would enhance the interpretability and transparency of the findings, especially given the within-subject design.

Daily Sample Size Reporting

Echoing Reviewer 1’s comment, I too found the reporting of the number of participants recruited on each specific day to be puzzling. While the authors offer an explanation, it still seems unnecessary and may distract from the main narrative. I suggest removing this detail or, if retained, relocating it to a footnote or appendix.

Justification for Within-Subject Design

The manuscript currently includes justification for the within-subject design in two separate locations. While the explanation is sound, it may be unnecessarily lengthy and slightly repetitive. I understand that this likely responds to Reviewer 1’s request for elaboration, but a more concise explanation in a single location would suffice. Moreover, the authors emphasize the limitations of the design, but could also briefly mention its advantages—for example, enhanced statistical power and control for individual differences.

Power Analysis

I suggest that the authors briefly address the absence of a priori power analysis. Although the within-subject design improves power, it would be helpful to explain why no power calculation was conducted during the planning stage.

Durability of the Effect

One additional point relates to the observation that the effects of the intervention persisted three days after exposure. While this is mentioned in both the abstract and results, the discussion section does not elaborate on the theoretical or practical implications of this finding. What does it mean, theoretically or in applied terms, that such effects last for three days? Moreover, I would caution against characterizing this as a "lasting" or "persistent" effect without qualification. Measuring the outcomes again only three days after the intervention offers a rather limited window into durability. It would be valuable for the authors to acknowledge this limitation more explicitly and, if possible, to suggest future research directions that examine longer-term impact. Even a brief discussion of these considerations would help contextualize the findings more appropriately

7. PLOS authors have the option to publish the peer review history of their article (what does this mean? ). If published, this will include your full peer review and any attached files.

**Do you want your identity to be public for this peer review?** For information about this choice, including consent withdrawal, please see our Privacy Policy .

Reviewer #1: **Yes: ** Dr. Assaf Suberry

Reviewer #3: No

---

## [Author Response · Author response to Decision Letter 2]

22 Aug 2025

Response to reviewers (second round) - see also our submitted word doc "Response to Reviewers".

We thank both reviewers for their time and their constructive and valuable feedback. We respond to their comments below.

Comments from Reviewer #1

Comment 1: On page 5, the authors state: “The individual pictured in Fig 1 has provided written informed consent (as outlined in PLOS consent form) to publish their image alongside the manuscript.” I recommend revising this for clarity and consistency: “The individual pictured in Figure 1 has provided written informed consent (as outlined in the PLOSOne consent form) to publish his image alongside the manuscript.”

Response: We were advised by Plos One staff to include this sentence, so we feel we should stick to the wording provided to us.

Comment 2: As noted in the first round of review, please consider including the main results in the abstract. You have added the sample size, which is helpful, but including the key outcomes and p-values would be more beneficial for readers who are screening the abstract of your manuscript.

Response: We do already mention that the key outcomes were fusion, trust and openness to friendship. However, we don’t have enough space in the abstract to explain the index we built using these three single outcome variables in an appropriate manner, so including p-values for index results would potentially cause confusion. Also, we prefer not to report p-values for the single index components as this is not what we pre-registered.

Comments from Reviewer #3

Introduction – Background on the War: The opening paragraph, which provides a chronological overview of recent events in the Israel-Gaza war, could be significantly shortened. I am uncertain about the relevance of this historical background for the reader, who is likely already familiar with the context. Additionally, although the authors make a commendable effort to present the events in a neutral and objective tone, some readers who strongly identify with either the Israeli or Palestinian side may perceive the description as biased or unfair. Given the ongoing nature of the war and the emotional sensitivity of the topic, this paragraph risks alienating readers and undermining trust in the research. I recommend either substantially condensing the paragraph or removing it altogether, and beginning the introduction with the theoretical background—which, in my view, should be the core of the introduction.

Response: We agree that discussing the background of the conflict is an extremely challenging task and also that the introductory text could be shortened. In the revised version, we have deleted the second paragraph discussing divisions within the US over the Israeli-Palestinian conflict. However, we would like to retain the first paragraph because we think that giving a short description on the reactions of the relevant international institutions to the conflict is the most even-handed approach. Many readers who are emotionally impacted by the conflict could perceive as biased and unfair a version of the article that does not even mention well-documented allegations of war crimes and crimes against humanity levelled at both sides. We are careful to refer only to official processes within internationally recognized institutions without making any value judgements on these processes. We have also deleted some remaining adjectives in the first paragraph that could possibly be interpreted as value laden.

Reporting Differences Across Outcome Dimensions: While I understand the rationale for creating a composite outcome index from identity fusion, trust, and openness to friendship (and I acknowledge the high intercorrelations reported in the appendix), the three variables represent conceptually distinct constructs. It would be valuable to examine whether the speech had different effects across these dimensions. If the effects were similar, the authors could briefly note this in the text and include the detailed results in the appendix. Either way, some comment in the main text would help clarify the distinctiveness (or lack thereof) of the outcome components.

Response: We note at the very end of sections “Study 1 – Results” and “Study 2” that we discuss the effects on the single components in the Appendix (Appendix S3 for study 1 and Appendix S4 for study 2). In both studies, the effects on all three components of the outgroup index were significant. At the end of Appendices S3 and S4 we now also state the effects on all three components of the outgroup index as shares of the pre-treatment gap of the ingroup value of the component and the outgroup value of the component to make the three effects comparable. In both studies we found that the effect on fusion was significantly larger than the effect on trust and the effect on openness to friendship. These results are theoretically plausible since fusion theory predicts primarily the treatment effect on fusion. Moreover, fusion with and a resulting sense of caring for the outgroup can plausibly exist in the context of a complex conflict without equally strong levels of trust or even openness to friendship. However, we also used different scales to measure fusion (from 1-5) and the other two outcomes (from 0-100) and the smallest movement on these scales is larger for the fusion scale which could also explain the observed patterns. Therefore, we would not give these results too much weight.

Effect Size Reporting: The results section would benefit from more explicit reporting of effect sizes, such as Cohen’s d. While the authors describe mean changes and provide p-values, including standardized effect sizes would enhance the interpretability and transparency of the findings, especially given the within-subject design.

Response: We now also report Cohen’s dz.

Daily Sample Size Reporting: Echoing Reviewer 1’s comment, I too found the reporting of the number of participants recruited on each specific day to be puzzling. While the authors offer an explanation, it still seems unnecessary and may distract from the main narrative. I suggest removing this detail or, if retained, relocating it to a footnote or appendix.

Response: We moved this detail to Appendix S2 and no longer discuss it in the main text.

Justification for Within-Subject Design: The manuscript currently includes justification for the within-subject design in two separate locations. While the explanation is sound, it may be unnecessarily lengthy and slightly repetitive. I understand that this likely responds to Reviewer 1’s request for elaboration, but a more concise explanation in a single location would suffice. Moreover, the authors emphasize the limitations of the design, but could also briefly mention its advantages—for example, enhanced statistical power and control for individual differences.

Response: We have deleted the second superfluous justification of the within-subject design in section “Study 1 - Design and hypotheses”. We discuss advantages of within-subject designs in the “Present research” section.

Power Analysis: I suggest that the authors briefly address the absence of a priori power analysis. Although the within-subject design improves power, it would be helpful to explain why no power calculation was conducted during the planning stage.

Response: We have added the following text in the first paragraph of section “Study 1 – Data”: “We did not conduct an a-priori power analysis because it seemed unlikely that we would be able to recruit the full n=400 sample due to the limited number of available Jewish and Muslim participants registered on Prolific. Also, we considered n=400 to be a sufficient upper bound given our prior experiences with within-subject designs.”

Durability of the Effect: One additional point relates to the observation that the effects of the intervention persisted three days after exposure. While this is mentioned in both the abstract and results, the discussion section does not elaborate on the theoretical or practical implications of this finding. What does it mean, theoretically or in applied terms, that such effects last for three days? Moreover, I would caution against characterizing this as a "lasting" or "persistent" effect without qualification. Measuring the outcomes again only three days after the intervention offers a rather limited window into durability. It would be valuable for the authors to acknowledge this limitation more explicitly and, if possible, to suggest future research directions that examine longer-term impact. Even a brief discussion of these considerations would help contextualize the findings more appropriately.

Response: We have added a paragraph in the “Discussion” section on this (the sixth paragraph starting with “A third limitation…”). We argue that partially persistent effects of a single exposure to a 3-minute speech in video format provides evidence for the effectiveness of the approach given that participants were likely exposed to many real-world stimuli between waves in the heated and emotionally intense phase of the conflict in December 2023. We also argue that this result is consistent with identity fusion theory, and we underline the need for future research that analyses effects over longer time periods. We also recommend further research on the long-term effects of repeated exposure to interventions or exposure to even more intense interventions.

---

## [Editor Report · Decision Letter 2]

28 Aug 2025

Appeals to shared suffering in the context of the Israeli-Palestinian conflict

PONE-D-25-02752R2

Dear Dr. Reinhardt,

We’re pleased to inform you that your manuscript has been judged scientifically suitable for publication and will be formally accepted for publication once it meets all outstanding technical requirements.

Kind regards,

Gal Harpaz, Ph.D.

Academic Editor

PLOS ONE

Additional Editor Comments (optional):

Thank you for your careful and thoughtful revisions.

You have addressed the reviewers’ concerns thoroughly, and I am pleased to recommend your manuscript "Appeals to Shared Suffering in the Context of the Israeli-Palestinian Conflict" for publication in PLOS ONE.
---

## [Editor Report · Acceptance letter]

PONE-D-25-02752R2

PLOS ONE

Dear Dr. Reinhardt,

I'm pleased to inform you that your manuscript has been deemed suitable for publication in PLOS ONE. Congratulations! Your manuscript is now being handed over to our production team.

Kind regards,

on behalf of

Dr. Gal Harpaz

Academic Editor

PLOS ONE